# Water Footprint, Blue Water Scarcity, and Economic Water Productivity of Irrigated Crops in Peshawar Basin, Pakistan

Tariq Khan [1,2,*], Hamideh Nouri [3], Martijn J. Booij [4], Arjen Y. Hoekstra [4,†], Hizbullah Khan [2] and Ihsan Ullah [5]

1. Department of Environmental Sciences, University of Haripur, Haripur 22620, Pakistan
2. Department of Environmental Sciences, University of Peshawar, Peshawar 25120, Pakistan; hizbullah@uop.edu.pk
3. Division of Agronomy, University of Göttingen, Von-Siebold-Strasse 8, 37075 Göttingen, Germany; hamideh.nouri@uni-goettingen.de
4. Department of Water Engineering and Management, Faculty of Engineering Technology, University of Twente, 7500 AE Enschede, The Netherlands; m.j.booij@utwente.nl (M.J.B.); a.y.hoekstra@utwente.nl (A.Y.H.)
5. Department of Geography, University of Peshawar, Peshawar 25120, Pakistan; ihsanullah@uop.edu.pk
* Correspondence: tariqkhan@uoh.edu.pk
† Deceased.

**Abstract:** Pakistan possesses the fourth largest irrigation network in the world, serving 20.2 million hectares of cultivated land. With an increasing irrigated area, Pakistan is short of freshwater resources and faces severe water scarcity and food security challenges. This is the first comprehensive study on the water footprint (WF) of crop production in Peshawar Basin. WF is defined as the volume of freshwater required to produce goods and services. In this study, we assessed the blue and green water footprints (WFs) and annual blue and green water consumption of major crops (maize, rice, tobacco, wheat, barley, sugarcane, and sugar beet) in Peshawar Basin, Pakistan. The Global Water Footprint Assessment Standard (GWFAS) and AquaCrop model were used to model the daily WF of each crop from 1986 to 2015. In addition, the blue water scarcity, in the context of available surface water, and economic water productivity (EWP) of these crops were assessed. The 30 year average blue and green WFs of major crops revealed that maize had the highest blue and green WFs (7077 and 2744 m$^3$/ton, respectively) and sugarcane had the lowest blue and green WFs (174 and 45 m$^3$/ton, respectively). The average annual consumption of blue water by major crops in the basin was 1.9 billion m$^3$, where 67% was used for sugarcane and maize, covering 48% of the cropland. The average annual consumption of green water was 1.0 billion m$^3$, where 68% was used for wheat and sugarcane, covering 67% of the cropland. The WFs of all crops exceeded the global average. The results showed that annually the basin is supplied with 30 billion m$^3$ of freshwater. Annually, 3 billion m$^3$ of freshwater leaves the basin unutilized. The average annual blue water consumption by major crops is 31% of the total available surface water (6 billion m$^3$) in the basin. Tobacco and sugar beet had the highest blue and green EWP while wheat and maize had the lowest. The findings of this study can help the water management authorities in formulating a comprehensive policy for efficient utilization of available water resources in Peshawar Basin.

**Keywords:** green water footprint; blue water footprint; canal irrigated crops; water scarcity; economic water productivity; Pakistan

## 1. Introduction

Currently, one-third of the human population lives in areas where water is scarce [1,2]. Freshwater is greatly threatened by human activities [3]. The overexploitation of water resources has put extra stress on this scarce resource, and this stress is increasing due to population growth, water pollution, and the impact of climate change [4–6]. Water

consumption for irrigation purposes has been estimated at around 70% of the total annual water withdrawal worldwide and continues to increase [7,8]. This growing demand for agricultural water has put additional pressure on the water supply for domestic and industrial activities [9]. Drought, aridity, and climate change threaten the food security of many countries around the globe [10]. Pakistan has serious challenges in achieving the food security and nutrition targets of the nation [11]. Although Pakistan is an agricultural country, where 70% of the population depends directly or indirectly on agriculture for their living [12], and has the world's fourth largest irrigation system [13,14], about 40% of the population faces food insecurity [15]. The irrigated area of Pakistan increased from 10.8 million hectares in 1961 to 20.2 million hectares in 2018 [16,17]. The agricultural sector uses about 69% of the available water resources as the major consumer, followed by industry and households with respective shares of 23% and 8% [12]. In Pakistan, the agricultural sector accounts for around 40% of the workforce, 22% of the national gross domestic product (GDP), and 65% of the rural population. This is why agriculture is considered to be the backbone of the country's economy [18]. Like many other developing countries, Pakistan has been facing water shortage problems in recent years. The country receives less rainfall than water that is lost to evaporation; the availability of surface water resources is decreasing, and if the situation continues, Pakistan will soon be a water-scarce country [19]. This is the utmost threat to sustainable food security [20], the job market, and the national economic state [14].

Metrics for water consumption and water scarcity have been developed over the past few decades. There are over 150 different indicators to measure water demand, water abstraction, water consumption, water stress, water productivity, and water scarcity in different sectors, especially agriculture [21–24]. Among them, the water footprint (WF) has the ability to illustrate the effects of human activities on limited water resources. WF is defined as the volume of freshwater required to produce goods and services. It is the consumptive use of water, and it is divided into green and blue water components. Green water is rainwater that seeps into the soil or is intercepted by crops and is partly consumed by evapotranspiration (ET); consumption of this water leads to the green WF. Blue water is fresh surface and groundwater, and the consumption of this water leads to the blue WF [25–27]. Blue and green water resources have different characteristics and, therefore, need different management strategies to optimize their productivity.

There are numerous WF studies at the global and regional levels using the water footprint assessment framework, which divides water resources into two major components, i.e., blue and green water for different products, services, individuals, or communities and geographical regions [28–32]. For instance, a study on water consumption by global wheat production during the period 1996–2005 showed that the Indus and Ganges river basins accounted for about 47% of the blue WF of wheat [33]. In another study, the global production and consumption of rice during the period 2000–2004 showed that the consumption of rice products in the European Union caused 2.3 billion $m^3$ of water evaporation in India, Thailand, USA, and Pakistan [34]. In a regional level study in Northeast China, the WF of maize production concerning climatic conditions, soil quality, and irrigation facilities was assessed [35]. In Indonesia, the WF concept was used in estimating interprovincial virtual water flow related to the trade of crop products [36]. Further, WF standards were also used to assess the economic water and land productivity related to crop production in Tunisia [37]. In Korea, WF of 42 agricultural products and 3 livestock products were studied from 2003 to 2012 [38]. In Iran, a comprehensive study assessed the green and blue WFs of 26 crops across 30 provinces from 1980 to 2010 for food security purposes [39]. In Pakistan, a first-ever study on the WF of crops showed that maize, barley, sugarcane, and sugar beet have a higher WF than the global average and the country needs to reduce the WF of crops for sustainable agricultural production [40]. However, on a basin level there are only a few studies, especially in arid to semi-arid regions. An example is a study for the upper Litani Basin in Lebanon, where the combined effect of soil mulching and drip irrigation was studied to reduce water scarcity problems during the period 2009–2016 [24,29]. In the

Heihe River basin in Northwest China, the blue and green WFs of agricultural, domestic, and industrial sectors for the period of 2004–2006 were determined [41]. In the Yellow River basin in China, the intra- and inter-annual variation of the WF of crops and blue water scarcity during the period 1961–2009 was assessed [42]. In Gauteng, South Africa, the WF of production of some vegetables including carrots, cabbage, beetroot, broccoli, and lettuce and their supply chain from farmers to consumers was estimated [43]. In Southern Amazonia, Brazil, green and blue WFs of agriculture (crop and pasture) in recent years (2000 and 2014) and future years (2030 and 2050) were modelled to study the impact of agricultural intensifications on crop water use [44]. In the Indus Basin, Pakistan, the WF of 17 million hectares of irrigated croplands in high spatiotemporal resolution was evaluated to support policymakers with a better vision of the national water and food security agenda [45]. In Punjab, Pakistan, blue, green, and grey WFs of cotton from production to textile were estimated to throw light on the WF of the textile industry [46].

Previous studies in Pakistan mostly did not incorporate local data into the modelling process of water productivity of crop production. To the best of our knowledge, this study is the first to feed local data to the crop WF assessment to simulate farming conditions that are close to reality. In addition, this research assesses blue water scarcity and the EWP of crop production in the basin to enhance the awareness of water scarcity and alleviate water-related food security challenges of the basin by improving food and cash production of the farming system.

The current study aimed to assess the water scarcity and crop water productivity in an arid and semi-arid basin, Peshawar Basin in Pakistan. The specific objectives of the study were (1) to estimate the spatial distribution of the blue and green WFs and the temporal variation in blue and green water consumption of major crops with locally collected data over the period 1986–2015; (2) to assess the pressure on the available surface water resources due to the water consumption by major crops; and (3) to determine the economic water productivity (EWP) of major crops in Peshawar Basin. The findings of this study could be used for efficient and sustainable crop production, better water resources management, and long-term planning for the food and water security of Pakistan.

## 2. Materials and Methods

### 2.1. Study Area

Peshawar Basin with an area of 5617 km$^2$ is located in an arid to semi-arid region, between longitudes 71.25 and 72.75 E longitudes and 33.75 to 34.50 N latitudes in the Khyber Pakhtunkhwa province of Pakistan [47]. The basin has 9.78 million inhabitants [48]. The main rivers supplying water to the basin are the Kabul River and Swat River. A substantial monsoon rainfall occurs from July to September [49,50]. Around 100 irrigation canals divert from these rivers with an estimated length of 290 km [51]. January is the coldest month with an average temperature of 4.4 °C and June is the warmest month with an average temperature 40.39 °C. The average annual rainfall is 600 mm. The basin consists of 60% cropland, 27% natural vegetation, 10% settlements, 1% water bodies, and 1% barren land [52]. Records from 1986 to 2015 have revealed that from the 60% cropland, 97% is covered by wheat, maize, sugarcane, tobacco, barley, rice, and sugar. Of this 97% cropland, wheat covers 43%, maize 24%, sugarcane 24%, tobacco 4%, barley 2%, sugar beet 1%, and rice 1% of the area [48] (Figure 1).

### 2.2. Data

Only two meteorological stations are located in Peshawar Basin (Figure 1). Precipitation, minimum and maximum temperature, wind speed, and solar radiation data were acquired from the regional office of the Pakistan Meteorological Department. Reference evapotranspiration ($ET_o$) was calculated according to the Penman–Monteith equation using daily solar radiation, wind speed, and maximum and minimum temperature and humidity [53].

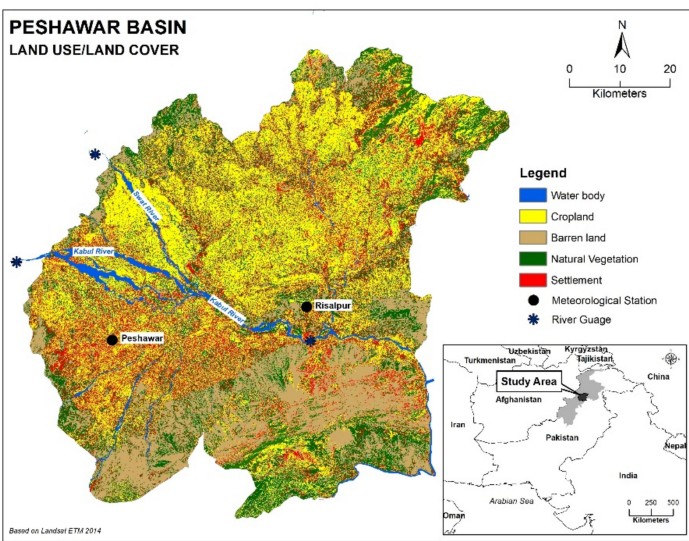

**Figure 1.** Land use and land cover map of Peshawar Basin, Pakistan.

The cultivated area of crops and yield per hectare were acquired from the local crop reporting service offices [48]. Soil properties, including bulk density, organic matter, and salinity for different horizons, were extracted from the Harmonized World Soil Database [54]. Soil texture was determined using the hydraulic properties calculator of Saxton et al. [55], see Table 1.

The basin was divided into two sub-regions using the Thiessen polygon method [56]; Peshawar meteorological station covers 35% of the area and the Risalpur station covers 65% of the area. There are three soil types in the basin, namely calcisols (65%), cambisols (25%), and rock outcrop (10%) [52]. Based on the soil types and climate conditions, the basin was further divided into six sub-regions by overlaying soil and climate maps. These sub-regions are Peshawar calcisol (PCL) 25%, Peshawar cambisol (PCM) 10%, Peshawar rock outcrop (PRK) 0.1%, Risalpur calcisol (RCL) 39%, Risalpur cambisol (RCM) 15%, and Risalpur rock outcrop (RRK) 10%, as shown in Figure 2.

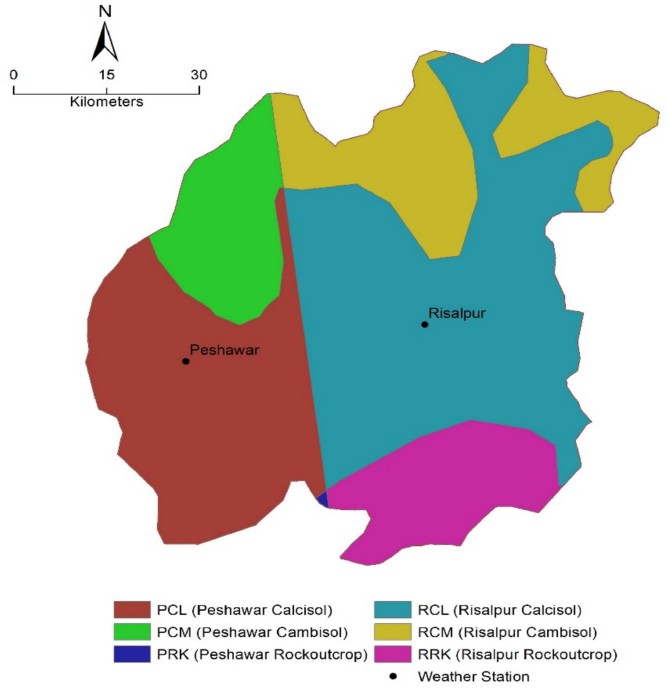

**Figure 2.** Sub-regions of Peshawar Basin with overlayed soil and climate maps.

Table 1. Soil characteristics of Peshawar Basin, Pakistan. PWP stands for permanent wilting point, FC for field capacity, SAT for saturation, and $K_{sat}$ for saturated hydraulic conductivity.

| Soil Type | Horizons | Texture USDA | Thickness (m) | Sand Fraction (%) | Silt Fraction (%) | Clay Fraction (%) | Bulk Density (kg/dm$^3$) | Organic Matter (wt. %) | Salinity (ds/m) | Stoniness (%) | Soil Water | | | |
|---|---|---|---|---|---|---|---|---|---|---|---|---|---|---|
| | | | | | | | | | | | PWP | FC | SAT | $K_{sat}$ |
| | | | | | | | | | | | (Volume%) | | | (mm/Day) |
| **Calcisols** | **Topsoil** | **Loam** | **0.3** | **39** | **40** | **21** | **1.32** | **0.7** | **1.6** | **4** | **13.5** | 27 | 46 | 196.5 |
| | Subsoil | Loam | 0.7 | 36 | 40 | 24 | 1.42 | 0.29 | 1.6 | 3 | 15 | 29 | 41 | 131.5 |
| Cambisols | Topsoil | Loam | 0.3 | 42 | 36 | 22 | 1.37 | 1 | 0.1 | 9 | 14 | 27 | 42 | 100 |
| | Subsoil | Loam | 0.7 | 40 | 35 | 25 | 1.39 | 0.4 | 0.1 | 12 | 15.3 | 28 | 41 | 116 |
| Rock Outcrop | Topsoil | Loam | 0.3 | 43 | 34 | 23 | 1.3 | 1.4 | 0.1 | 26 | 14.7 | 28 | 43 | 151.2 |
| | Subsoil | Clay loam | 0.7 | 42 | 30 | 28 | 1.37 | 0.3 | 0.7 | 3 | 17.1 | 29 | 41 | 4.37 |

The spatial distribution of crops in each sub-region was extracted from satellite images; Sentinel-2 MSI images (10 m spatial resolution) were downloaded from the Copernicus Open Access Hub [52]. The crops maturity stages were acquired from the Khyber Pakhtunkhwa's crop calendar [57]. The dates of these images were selected according to the crop maturity dates, for example, for sugarcane, the date of the image was 9 September 2018, for wheat it was 15 March 2018, and for maize it was 8 July 2018. The downloaded images were stacked, mosaicked, and cropped for the study area. Each subset image was classified by the supervised classification maximum likelihood method for the extraction of the major crop area. The field management data, i.e., application of fertilizers and irrigation schedule, were acquired from the national fertilizer development center and local irrigation departments [48]. Irrigation efficiency, irrigation depths, and surface wetted area per irrigation treatment were taken from reference [58].

### 2.3. Methods
#### 2.3.1. Estimation of Blue and Green Water Footprints of Major Crops in Peshawar Basin

The annual blue and green WFs of major crops in Peshawar Basin were estimated using the Global Water Footprint Assessment Standards [24]. FAO's AquaCrop model (version 6.1) was used to simulate the soil–water balance and crop yield [59,60]. The model estimates ET and crop yield by simulating the dynamic soil–water balance (Equation (1)) and the biomass growth on a daily basis.

$$S_i = R_i + I_i + CR - RO_i - Dr - ET_i \tag{1}$$

where $S$ is soil water content (mm) on day $i$, $R$ is rainfall (mm), $I$ is irrigation (mm), $CR$ is the capillary rise (mm), $RO$ is a surface runoff, $Dr$ is drainage to the saturated zone (mm), and $ET$ is evapotranspiration [24].

The AquaCrop's default crop characteristics were revised based on local data or the available literature and are presented in Table 2 [61].

**Table 2.** Crop characteristics of Peshawar Basin, Pakistan.

| Crop | Maturation Date | Covered Area (10³ Hectares) [48] | Harvest Index (%) |
|---|---|---|---|
| Wheat | 1st November | 142.63 | 45 |
| Maize | 15th May | 81.34 | 48 |
| Sugarcane | 1st November | 79.64 | 75 [48] |
| Rice | 15th July | 2.00 | 43 |
| Tobacco | 1st February | 13.09 | 85 |
| Barely | 1st November | 7.72 | 33 |
| Sugar beet | 15th October | 6.65 | 54 |

The model was calibrated following the FAO guidelines [62]. The parameters such as soil fertility and biomass production were adjusted, and the model was run several times until the simulated yield closely agreed with the observed yield [48]. This procedure was performed for each crop except sugarcane since the model did not perform well for sugarcane; hence, AquaCrop's simulated ET and observed yield for sugarcane for the period 1986–2015 was used instead. The root mean square error (RMSE) was used as an indicator to evaluate model performance. The RMSE percentage of wheat was 6.3%, maize 26.8%, rice 6.6%, tobacco 6.9%, barley 3.6%, and sugar beet 15%. The model was run for all sub-regions for 30 successive one-year periods (1986–2015) except for PRK and RRK, since rock layers in these regions prevent farming activities [63].

The output of the AquaCrop simulations was divided into blue and green parts using the method introduced by Chukalla et al. [64]. The blue and green components of crop

water use (CWU) were calculated by adding blue and green *ET* over the period of crop growth (Equations (2) and (3)):

$$CWU_b = \sum_{t=1}^{T} \frac{S_{bt}}{S_t} ET_t \times 10 \tag{2}$$

$$CWU_g = \sum_{t=1}^{T} \frac{S_{gt}}{S_t} ET_t \times 10 \tag{3}$$

where $CWU_b$ and $CWU_g$ are blue and green water consumption (m$^3$), $S_{bt}$ and $S_{gt}$ are changes in blue and green soil water storage over the growing period, and 10 is the conversion factor from mm to m$^3$. The $WF_b$ and $WF_g$ were obtained by dividing $CWU$ by the crop yield (ton) $Y$ using Equations (4) and (5) [26].

$$WF_b = \frac{CWU_b}{Y} \tag{4}$$

$$WF_g = \frac{CWU_g}{Y} \tag{5}$$

2.3.2. Blue Water Scarcity and Economic Water Productivity

The runoff records of Kabul and Swat Rivers from 1986 to 2015 were obtained from the Pakistan Water and Power Development Authority (WAPDA). Since there were no national or local standards for the environmental flow requirements (EFR) for the study area, 80% of the natural runoff was allocated as EFR following Nouri et al. [29]. The remaining 20% was considered as the blue surface water available ($WA_{blue}$) in the basin. The blue water scarcity ($WS_{blue}$) as a result of crop water consumption is defined as the ratio of the blue water footprint ($WF_{blue}$) of crops to the available blue surface water ($WA_{blue}$) from 1986 to 2015, as shown in Equation (6) [65].

$$WS_{blue} = \frac{\sum WF_{blue}(crop)}{WA_{blue}} \times 100 \tag{6}$$

Economic water productivity (USD/m$^3$) was calculated by multiplying water productivity (kg/m$^3$) by the price of crops (USD/kg) [37]. Since crops' prices fluctuated over the period 1986–2015, the average market prices for only 2019 were obtained by a questionnaire survey from four markets of the districts Peshawar, Charsadda, Mardan, and Nowshera cities located in the basin.

## 3. Results

### 3.1. Spatial Distribution of Blue and Green WFs of Major Crops in Sub-Regions of Peshawar Basin

The average annual blue and green WFs of crops from 1986 to 2015 varied among sub-regions. The annual average blue WFs of maize, rice, tobacco, wheat, barley, sugarcane, and sugar beet were 7077, 3932, 2176, 1913, 1561, 181, and 174 m$^3$/ton, and the annual average green WFs were 2744, 2254, 1985, 1535, 1603, 67, and 45 m$^3$/ton, respectively. The blue WFs of all crops exceeded the green WFs throughout the basin (Figure 3).

In all sub-regions, maize had the highest blue and green WFs while sugar beet had the lowest values.

There were substantial differences in the blue and green WFs of crops in the sub-regions. For instance, the blue WFs of maize were 5982 m$^3$/ton in PCL, 7100 m$^3$/ton in PCM, 7602 m$^3$/ton in RCL, and 7624 m$^3$/ton in RCM, while the green WFs were 2256 m$^3$/ton in PCL, 3093 m$^3$/ton in PCM, 2830 m$^3$/ton in RCL, and 2799 m$^3$/ton in RCM. The WF of cambisols was high as compared to that of calcisols, and the reason could be the high organic content in cambisols (Figure 4).

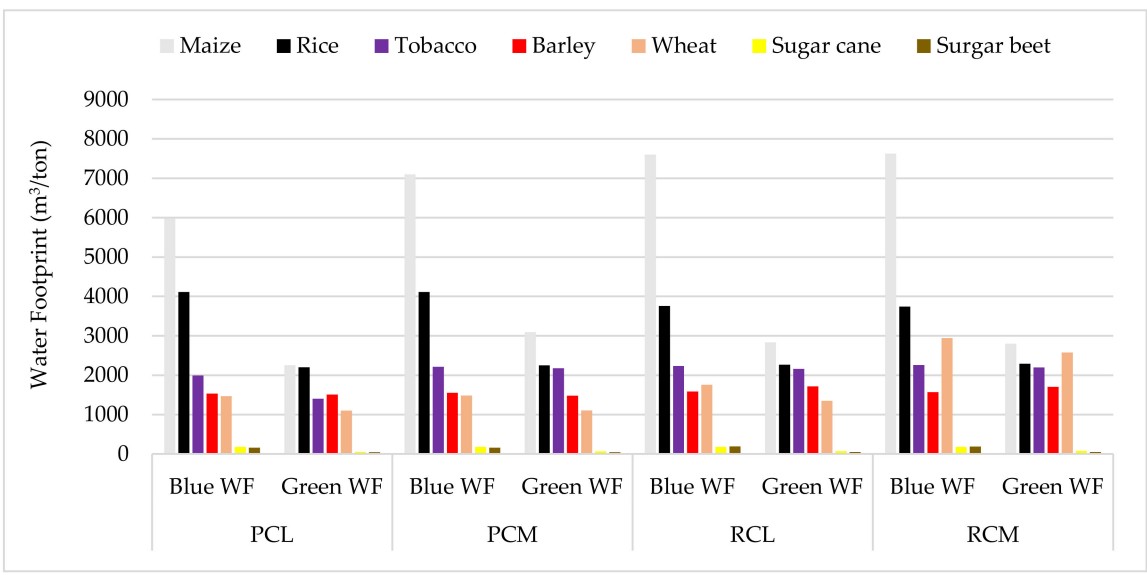

**Figure 3.** Annual blue and green water footprints of major crops in sub-regions of Peshawar Basin (1986–2015).

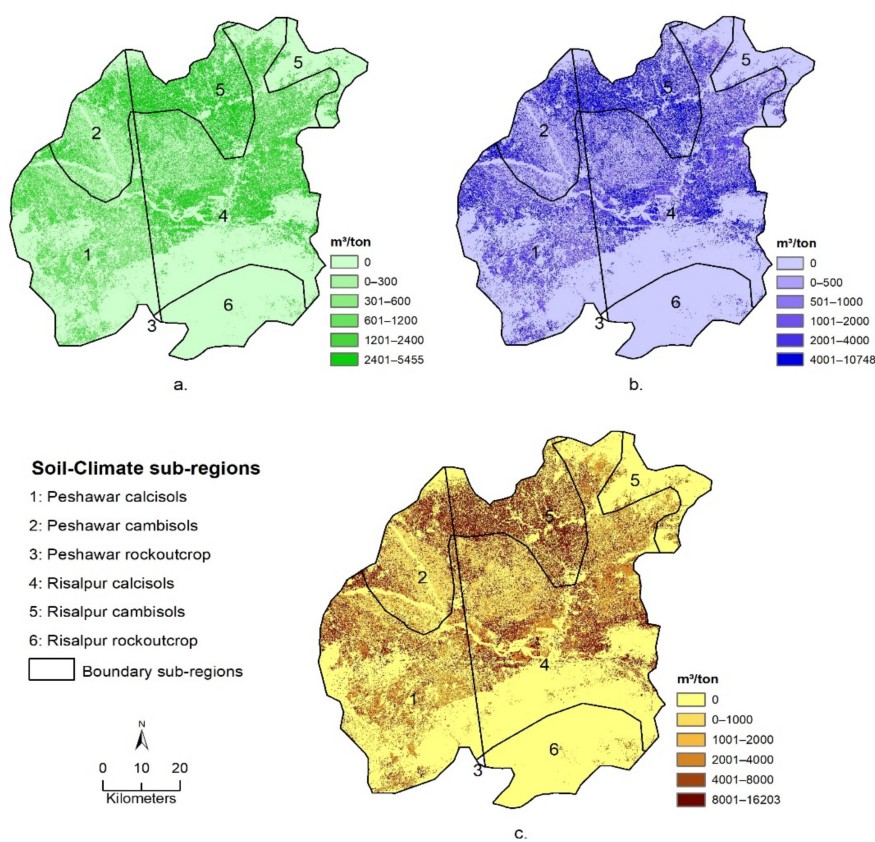

**Figure 4.** Spatial distribution of average annual green (**a**), blue (**b**), and total WF (**c**) of major crops in Peshawar Basin (1986–2015).

### 3.2. Temporal Variation in Blue and Green Water Consumption by Major Crops in Peshawar Basin

The annual blue and green water consumption by different crops changed over time. The average annual blue water consumption of sugarcane, maize, wheat, tobacco, sugar beet, rice, and barley was 655, 623, 494, 57, 32, 14, and 11 million m$^3$, while the average annual green water consumption was 308, 236, 391, 52, 8, 8, and 11 million m$^3$, respectively (Figure 5a,b). Although the blue and green WFs of rice were high (3932 and

2254 m³/ton, respectively) compared to the blue and green WFs of sugarcane (181 and 67 m³/ton, respectively), the blue and green water consumption values were comparable since sugarcane was cultivated on 24% of the cropland in the basin and rice on 1% of the cropland; therefore, the water consumption of sugarcane was high (655 million m³).

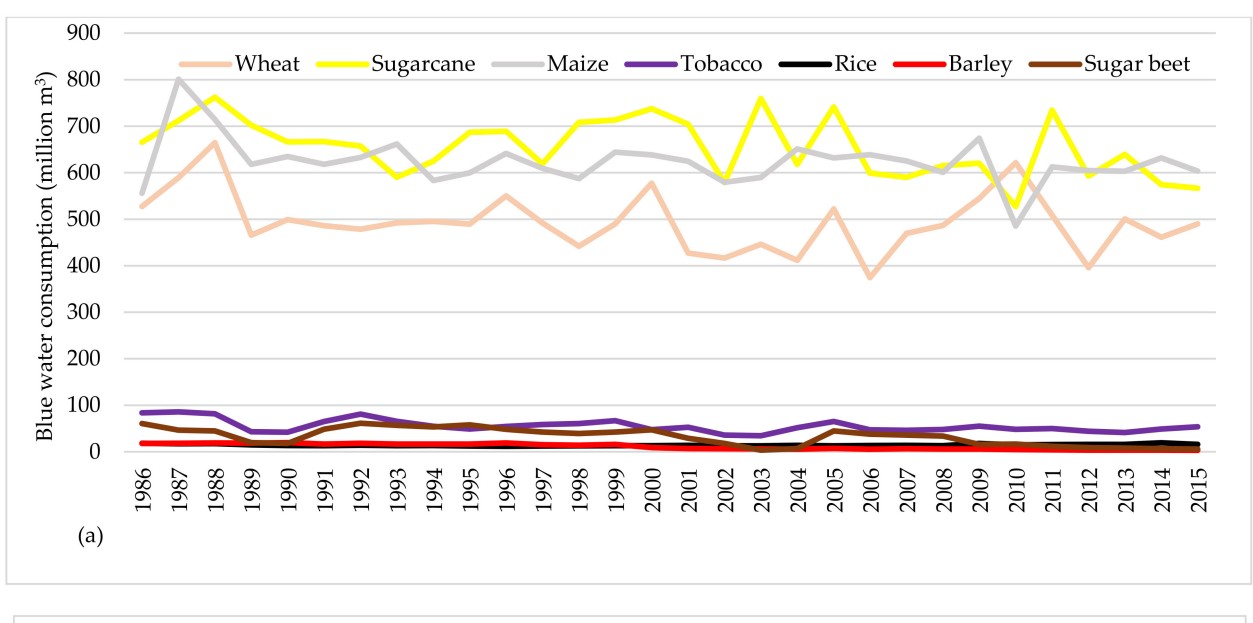

(a)

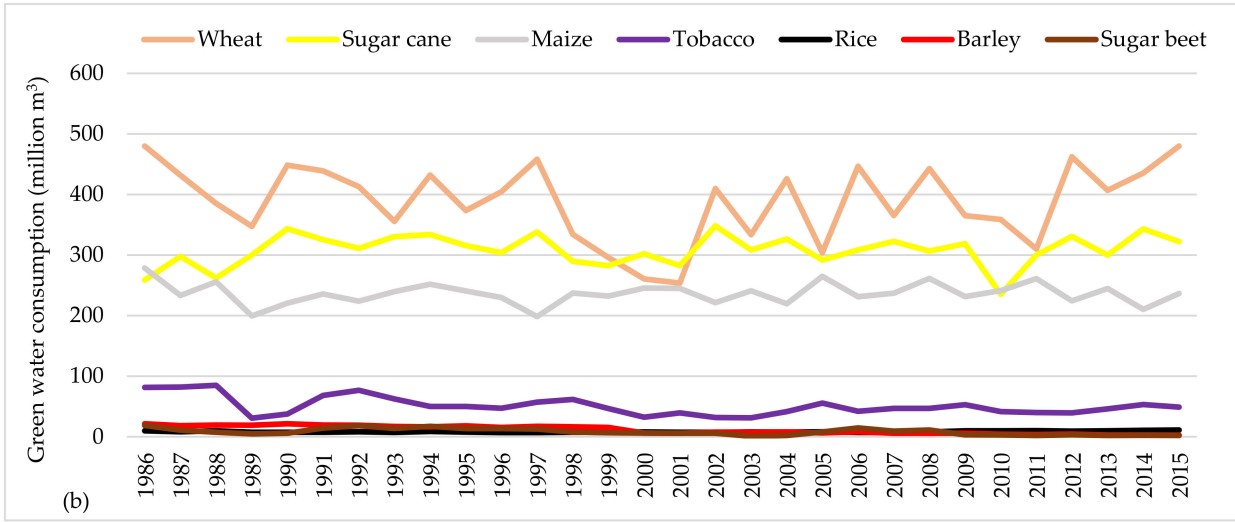

(b)

**Figure 5.** Variation in water consumption of major crops in Peshawar Basin (1986–2015), (**a**) blue water consumption (million m³) and (**b**) green water consumption (million m³).

The results showed that wheat, sugarcane, and maize consumed a high amount of blue and green water as compared to tobacco, rice, barley, and sugar beet during the period 1986–2015, (Figure 5a,b).

The thirty-year annual average blue and green water consumption values of major crops in Peshawar Basin were estimated as 1.9 billion m³ and 1.0 billion m³, respectively. The maximum blue water consumption was recorded in 1988 (2.3 billion m³) and the minimum in 2002 (1.6 billion m³), while the maximum green water consumption was recorded in 1986 (1.2 billion m³) and the minimum consumption was recorded in 2001 (0.83 billion m³), see Figure 6.

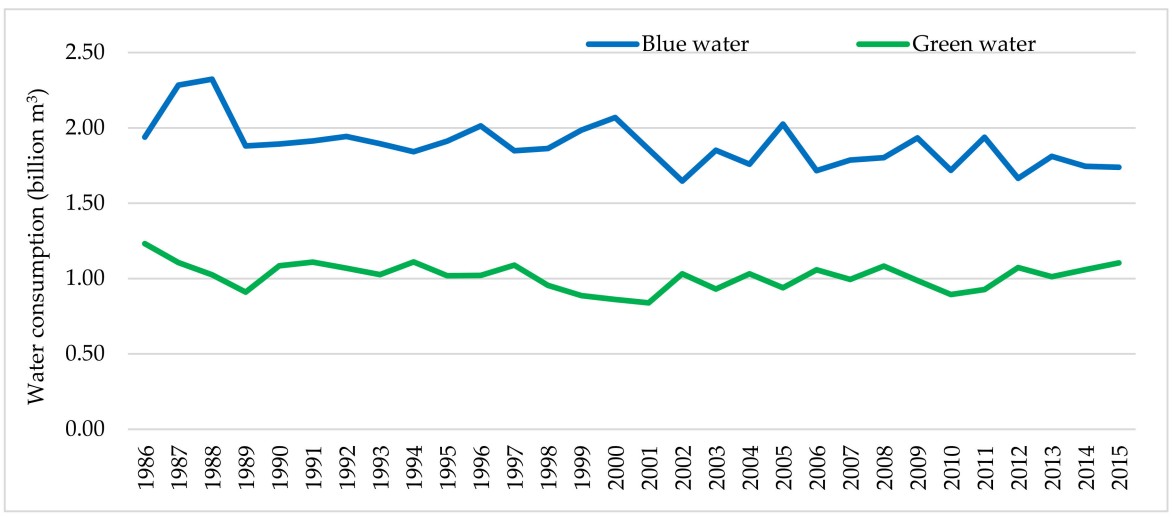

**Figure 6.** Variation in annual blue and green water consumption of major crops in Peshawar Basin.

*3.3. Blue Water Consumption versus Available Surface Water*

The thirty-year annual average discharge of Kabul and Swat Rivers was 18 billion m$^3$ and 12 billion m$^3$ per year, respectively (Figure 1). Hence, the thirty-year annual average total surface water inflow to the basin from these two sources was 30 billion m$^3$, while the outflow from the basin during the same period was approximately 27 billion m$^3$ (Figure 1). The 80% EFR was approximately 24 billion m$^3$ and the remaining 20% (6 billion m$^3$) was considered available surface water for various consumptive purposes (Figure 7).

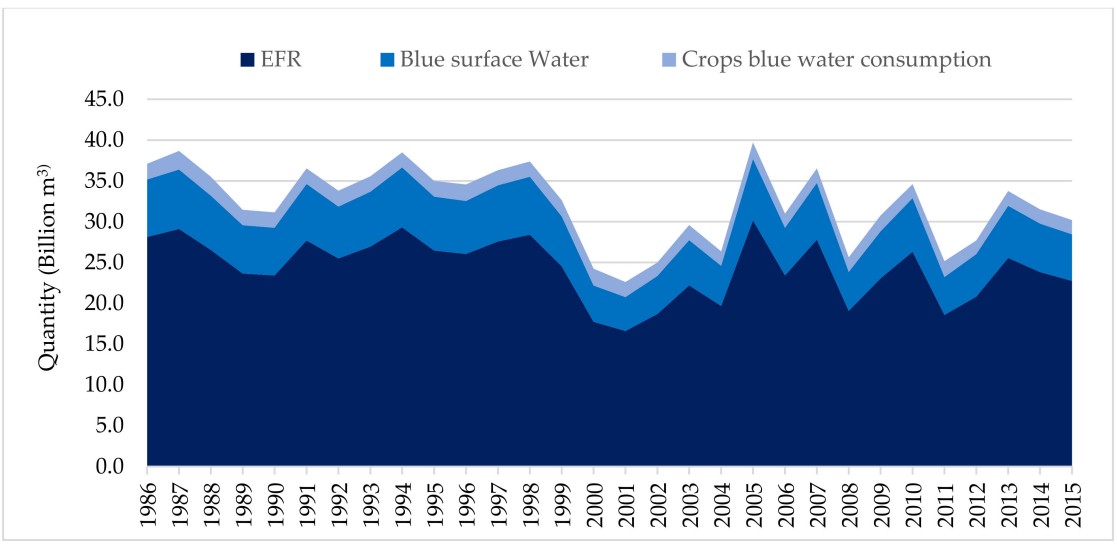

**Figure 7.** Contribution of EFR, non-utilized blue surface water, and crop water consumption in Peshawar Basin (1986–2015).

In this study, only irrigation from surface water was considered since groundwater irrigation is rare in the basin [48]. The annual average blue WF of crop production by surface water irrigation was estimated at 1.9 billion m$^3$, which is around 31% of the available surface water in the basin (6 billion m$^3$). The blue water scarcity over the period of 30 years varied from 25% in 1994 to 47% in 2000. The high scarcity recorded in 2000 was due to low flows in the river, see Figure 8. The thirty-year annual average scarcity was less than 50%, which was within sustainable limits and may be termed as low water scarcity [59] (Figure 8, Table 3).

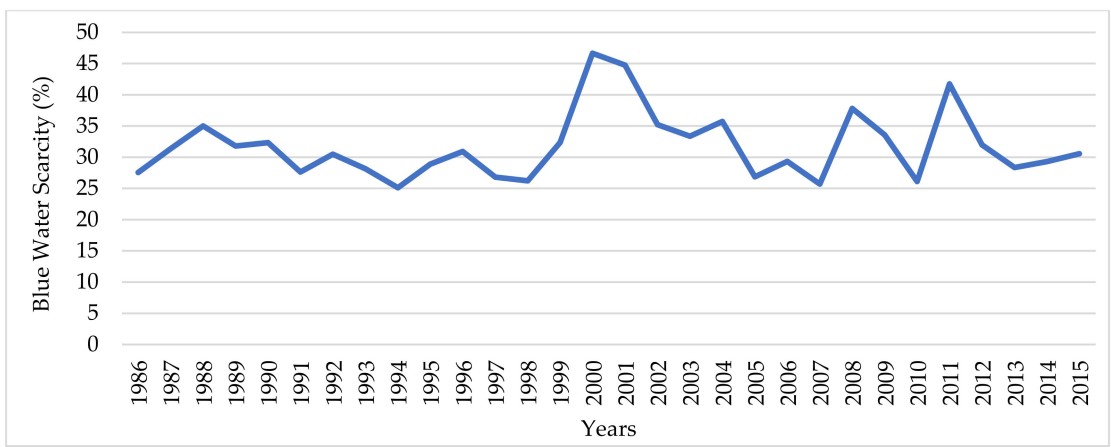

**Figure 8.** Blue water scarcity in Peshawar Basin (1986–2015).

**Table 3.** Classification of blue water scarcity [65].

| Water Scarcity Levels | Thresholds |
|---|---|
| Low water scarcity | <100% |
| Moderate water scarcity | 100–150% |
| Significant water scarcity | 150–200% |
| Severe water scarcity | >200% |

*3.4. Economic Water Productivity*

The blue and green economic water productivity (EWP) of major crops in Peshawar Basin is presented in Figure 9. On average, tobacco had the highest blue EWP (0.58 USD/m$^3$) followed by barley (0.26 USD/m$^3$), while maize had the lowest blue EWP (0.03 USD/m$^3$). Similarly, the green EWP of tobacco was the highest (0.65 USD/m$^3$) followed by sugar beet (0.57 USD/m$^3$), sugarcane (0.49 USD/m$^3$), barley (0.26 USD/m$^3$), and rice (0.24 USD/m$^3$). The high EWP of tobacco and barley was due to the high price and low WFs of these crops, and similarly, the low EWP of maize was because of its high WF and low price in the basin.

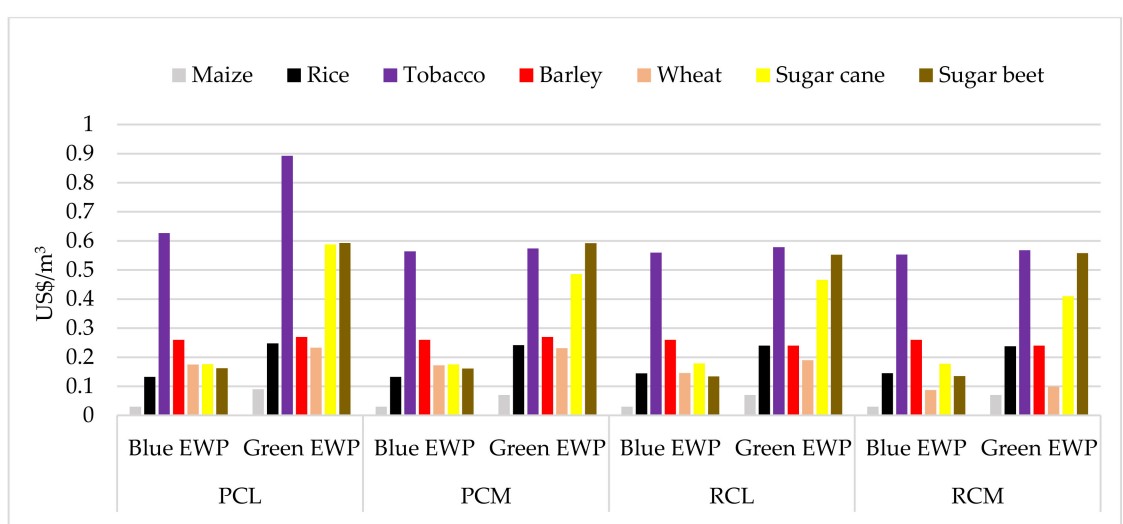

**Figure 9.** EWP of major crops in different sub-regions of Peshawar Basin.

The blue and green EWP of maize and rice were similar in all sub-regions, except PCL. The blue and green EWP of tobacco slightly varied in different sub-regions but was highest

in all sub-regions. Barley had the same blue EWP in all sub-regions, while the green EWP was high in PCL and PCM as compared to that in RCL and RCM. Wheat had the lowest blue and green EWP in RCM as compared to that in PCL, PCM, and RCL. Sugarcane had a similar blue EWP in different sub-regions, but different green EWPs. The green EWP of sugar beet was similar in PCL, PCM, RCL, and RCM sub-regions, while the blue EWP differed slightly in different sub-regions. In general, the results showed that the blue and green EWP of the soil type calcisol was relatively high as compared to that of the soil type cambisol since crops on cambisol have a higher WF than do those on calcisols.

## 4. Discussion

### 4.1. Comparison with Literature

The results indicate that the WF of all crops in Peshawar Basin, except barely and sugarcane, was very high compared to the global average (Figure 10) [66]. In general, maize, rice, and tobacco had the highest WFs of the seven crops. Although there are hundreds of WF studies on different crops in different spatial extents and geographical locations, there are only a few studies for comparison in Pakistan; those studies were conducted for of the whole country and not specifically on Peshawar Basin. Mekonnen and Hoekstra reported the global WF of crops and derived crop products including Pakistan [66], and Ghufran et al. reported the WF of cereals and selected minor crops of Pakistan [40]. Our results showed higher values of the blue and green WFs for all crops compared to the national rate reported by two mentioned studies except for sugarcane and sugar beet, which have low green WF values [40,66]. Ghufran et al. did not differentiate between blue and green WFs, and all our results were high compared with their combined WFs except for sugarcane and sugar beet [40]. The reasons for these major differences could be the methods used to estimate *ET*, input data, different modelling processes, and the scope of the studies.

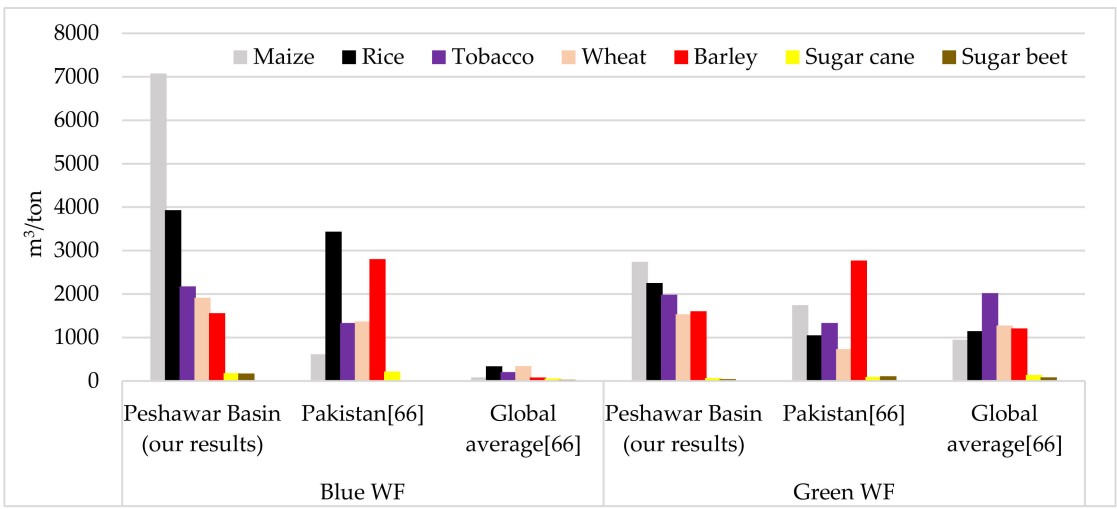

**Figure 10.** Comparison of the blue and green WFs of major crops in Peshawar Basin with those of the rest of Pakistan and the global average [66].

In this study, $ET_o$ was calculated according to the Penman–Monteith equation and using the AquaCrop model to estimate *ET*. In the first study [66], a grid-based water balance model was used to estimate *ET*, while in the second study [40] the authors calculated $ET_o$ by the Penman–Monteith equation and used the CROPWAT model to estimate *ET*. In the current study, the input data, i.e., all climatic data, crop characteristics, and irrigation schedules collected from local departments, were used to simulate conditions closer to reality, while in both the previous studies, local conditions were not utilized; instead, remote sensing data were used. For instance, in [66] the climatic data were acquired from Climatic Research Unit-Timeseries (CRU-TS-2.1) and crop growing area and irrigated fraction were deduced from the MICRA 2000 database. The application of coarse resolution

CRU weather data for basin-scale and the static rate of MIRCA2000 introduces significant uncertainties to the results. In [40], the climatic data and irrigation schedule were extracted from the CLIMWAT 2.0 software, whereas soil types were not even considered in the simulation process. Further, both mentioned studies assessed WF of Pakistan as a whole and did not take into account diverse climatic and water variability across the country. A simple example of these variations can be seen in the precipitation rate; the average annual rainfall in Peshawar Basin was 600 mm while in the southern parts of Punjab, Balochistan, Khyber Pakhtunkhwa, and Sindh the annual rainfall was less than 250 mm [67].

This assumption of homogeneity across different climate zones may lead to multiple sources of errors and biases resulting in over- or under-estimation of WFs. Changes in the WF of cereal production in Saskatchewan, Canada was significantly affected by local precipitation [68]. A recent study in Iran reported different responses of 26 crops in five climatic zones of the country and demonstrated the noticeable changes in their WFs [39]. Another water-scarce country, Tunisia, reported significant differences in crops' WFs in different climates; this study highlighted regional differences of WFs of various crops [37]. A WF assessment study in Morocco reported various WFs in different river basins [69]. Assessing the driving mechanism of the WF of crop production in China demonstrated the significant impact of precipitation on optimizing WF [70]. All mentioned examples exhibit the importance and necessity of regional assessment of crops' WFs to reflect close-to-real world conditions. Our study in Peshawar Basin integrated local data and modelling capabilities to assess the WFs of major crops of the basin more accurately.

### 4.2. Limitations and Uncertainties

This study presents a comprehensive WF assessment of major crops in Peshawar Basin in Pakistan; however, there are some limitations, such as the unavailability of data and simulation of perennial crops by the AquaCrop model. Due to the unavailability of data regarding the crop market prices for the period 1986–2015, the market prices of 2019 were considered for estimating EWP. Since prices increase with time, for instance, the price of wheat in 1991 was 142.9 USD/ton and rose to 286.4 USD/ton in 2016 [71], the prices from 2019 were higher as compared with the 1986 prices. Therefore, the EWP values of 2019 were lower compared to those of the year 1986. Since AquaCrop simulated biomass for crops having only a single growth cycle [63] and did not perform well in simulating perennial crop such as sugarcane, the observed yield was used in estimating the blue and green WFs.

### 4.3. Practical Implications

The WF and EWP results indicate that for sustainable economic growth, appropriate agricultural practices considering the WF and EWP of crops must be adopted. To tackle the increasing demand for food using the available agricultural resources, sustainable agricultural management practices (e.g., new irrigation technologies, fertilization, crop distribution pattern, etc.) need to be locally evaluated and, when approved, introduced to farmers to increase water productivity. A small-scale trial in Pakistan evaluated the efficiency of sprinkler and trickle irrigation techniques and showed an efficiency of 75% to 80% compared with other surface irrigation methods [67]. However, their impacts on the WF have not been assessed yet. A recent study in Lebanon claimed that mulching and drip irrigation could reduce blue water WF of the basin by 5% [24,29]. Another study evaluated the impact of different nitrogen fertilizer rates on the blue and grey WF and declared up to a 40% reduction in WF by appropriate fertilization rates [72]. A regional study in China stated that a crop's WF is more impacted by agricultural management than inter-annual climate variability [73]. In a similar study, the effect of different agricultural management on green and blue WFs revealed that deficit irrigation could reduce blue WF by 38% while mulching cut it by 10% [74]. The impact of different irrigation systems on the wheat WF in China was evaluated; results showed that micro irrigation had a lower WF compared to that of furrow and sprinkler systems. Recent studies proved that cropping

pattern and cropping calendar have the potential to enhance water use efficiency, water productivity, and WF [29,75–78]. We recommend a comprehensive national study to assess all these possible alternatives of efficient water use to find sustainable strategies to enhance the water productivity and reduce the WF of crop production.

### 4.4. Recommendations

Due to the water scarcity issues faced by the country, it is highly recommended that a comprehensive irrigation policy should be formulated to gradually shift the irrigation practices into more water-efficient techniques that are socially acceptable and financially viable. Based on the demand, comprehensive long-term planning is needed to investigate a transition from water-intensive crops to low WF crops with high EWP. The water management authority is advised to adopt a proactive policy to efficiently utilize the 3 billion $m^3$/year of water that currently flows down from the basin unutilized. Further, research is needed to assess the impact of water-efficient irrigation techniques such as sprinkler or drip irrigation, the application of different fertilizers, and cropping pattern on the WF and EWP while socio-economic conditions of the region are not threatened.

### 5. Conclusions

To alleviate the water and food security challenges of Pakistan, this study assessed WF and EWP of the major crops produced in Peshawar Basin. Our results showed that among all major crops of the basin, maize, rice, and tobacco had the highest blue and green WFs compared to those of wheat, barley, sugarcane, and sugar beet. Interestingly, crops with lower WFs, such as sugarcane and wheat, consumed most of the blue and green water because of their large cultivated area. Sub-region PCL had the highest blue and green EWP values while RCM had the lowest ones due to differences in soil type and climatic variation. From an economic perspective, crops with a high EWP (e.g., tobacco and sugar beet) were cultivated on a small part of the cropland compared to crops with a low EWP and high water consumption (e.g., wheat and maize). On average, annual blue and green water consumption of major crops was estimated at 1.9 billion $m^3$ and 1.0 billion $m^3$, respectively. Annually, 3 billion $m^3$ of unutilized freshwater is lost from the basin. The findings of this study can help the water management authorities in formulating a comprehensive policy for efficient utilization of available water resources of Peshawar Basin.

**Author Contributions:** Conceptualization, T.K., A.Y.H., H.N., M.J.B. and H.K.; methodology, T.K., A.Y.H., H.N. and M.J.B.; software, T.K., A.Y.H., H.N. and M.J.B.; validation, T.K., A.Y.H., H.N., M.J.B. and H.K; formal analysis, T.K., A.Y.H., H.N., M.J.B., H.K. and I.U.; investigation, T.K., A.Y.H., H.N., M.J.B., H.K. and I.U.; resources, T.K., A.Y.H., H.N., M.J.B., H.K. and I.U.; data curation, T.K., A.Y.H., H.N., M.J.B. and I.U.; writing—original draft preparation, T.K and H.N.; writing—review and editing, T.K., H.N., M.J.B. and H.K.; visualization, T.K., H.N., M.J.B., H.K. and I.U.; supervision, A.Y.H., H.N., M.J.B. and H.K. All authors have read and agreed to the published version of the manuscript.

**Funding:** The research received no external funding.

**Institutional Review Board Statement:** Not applicable.

**Informed Consent Statement:** Not applicable.

**Acknowledgments:** It is a privilege to acknowledge with gratitude the generous financial support provided by the Higher Education Commission (HEC) of Pakistan under the International Research Support Initiative Program (IRSIP) fellowship at the Department of Water Engineering and Management, University of Twente, The Netherlands.

**Conflicts of Interest:** The authors declare no conflict of interest.

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
