# Peer review of "Water Footprint, Blue Water Scarcity, and Economic Water Productivity of Irrigated Crops in Peshawar Basin, Pakistan"

_water, doi:10.3390/w13091249_

Round 1

Reviewer 1 Report

Comments in the PDF attached

Author Response

The following responses have already been added to MS as track change.

Point 1: The abstract needs a brief conclusion and a takeaway message for the reader to improve the impact of the paper. The authors have discussed the key results in detail in the abstract section but have not provided brief concluding remarks.

Response 1:  The concluding remark "The finding of this study can help the water management authorities in formulating a comprehensive policy for efficient utilization of available water resources in Peshawar basin" is added to the abstract.

Point 2: The introduction should cover a detailed explanation of water footprint and a more detailed explanation of blue and green WF.

Response 2: The definition of WF is added to the introduction while the blue and green WF has already been explained.

Point 3: The literature study should be much more extensive. I think there is a lot of literature available on this topic and the authors should explore more. There are several new studies that need to be cited. Especially for crop water footprint in the Peshawar region.

Response 3: The cited literature has been organized into 1) WF global level studies, 2) WF regional level studies, 3) WF basin level studies. A total of 13 studies have been referenced i.e. reference number 28 to 40. Howevere, a limited studies have been found specific to Peshawar basin, which have been cited already.

Point 4: Please improve the presentation of Figure 1. Either enlarge the figure or replace it with a more visible map. The figure is not visible when printed. Also, it is difficult to read the text on the map.

Response 4: the old map has been replaced with a more enlarged and visible map (Figure 1) added to the text.

Reviewer 2 Report

The manuscript needs some minor corrections:

  • the "crop yield" value is given on line 195 - no units were given (please complete)
  • In Figure 3, the explanation of the abbreviations PCL, PCM, RCL, RCM would greatly facilitate the analysis of this graph
  • Fig. 5: In the description in the graph "green water consumption", the unit m3 is missing.

Author Response

Point 1: The "crop yield" value is given on line 195- no units were given (please complete.

Response 1:  "crop yield" unit i.e. "ton" was added on line 195.

Point 2: In Figure 3, the explanation of the abbreviations PCL, PCM, RCL, RCM would greatly facilitate the analysis of this graph.

Response 2: The explanation of the abbreviations PCL, PCM, RCL and RCM is given in lines 147 to 150 and is described in Figure 2.

Point 3: Fig.5: In the description in the graph "green water consumption" the unit m3 is missing.

Response 3: Unit "m3" was added to "green water consumption" in Figure 5.

Reviewer 3 Report

The manuscript describes the assessment of blue and green water footprint and water consumption of major crops (maize, rice, tobacco, wheat, barley, sugar cane and sugar beet) in the Peshawar basin in Pakistan. The results presented can help to introduce efficient, sustainable and economically-profitable strategies to improve the utilization of available freshwater resources and crop production and to alleviate the water and food security challenges of Pakistan. The topic is of current interest for both the scientific and public communities, as well as water management authorities and fits within the scope of the Journal.

The manuscript is written in correct English regarding grammar and style. I made some minor suggestions below, but English is not my first language so I recommend a revision by someone more competent.

Specific Comments and Suggestions:

P 2; L 99: please correct into „basin to enhance“ Ä‘

P 4; L 138: Saxton et al. [49]

P 5; L 176: correct into „and presented in Table 2“

P 5; L 180: correct into „closely agreed“

P 5; L 189: Chukalla et al. [58]

P 8; L: 251: billion m3, respectively

P 9; L 253-256: figure caption is wrongly placed in this part of the text

P 9; L 265: Y axis legend should be corrected (only the unit is indicated), e.g.  „Quantity (billion m3)“ or something more appropriate

P 11; L 309: correct into „compared to both studies“

P 11; L 327: correct into „EFR, the basin still has“

P 12; L 361: tobacco

Author Response

All the following corrections can be found in revised MS as track change

Point 1: P 2; L 99: please correct into "basin to enhance".

Response 1: corrected

Point 2: P 4; L 138: Saxton et al. [49].

Response 2: corrected

Point 3: P 5; L 176: correct into "and presented in Table 2".

Response 3: corrected

Point 4: P 5; L 189: Chukalla et al. [58].

Response 4: corrected

Point 5: P 8; L 251: billion m3, respectively.

Response 5: corrected

Point 6: P 9; L 253-256: figure caption is wrongly placed in this of the text

Response 6: corrected

Point 7: P 9; L 265: Y-axis legend should be corrected (only the unit is indicated) e.g. "Quantity (billion m3)" or something more appropriate.

Response 7: Quantity added to Y-axis legend

Point 8: P 11; L 309: correct into "compared to both studies"

Response 8: corrected

Point 9: P 11; L 327: correct into "EFR, the basin still has".

Response 9: corrected

Point 10: P 12: L 361: tobacco

Response 10: corrected